# The Effects of Berry Polyphenols on the Gut Microbiota and Blood Pressure: A Systematic Review of Randomized Clinical Trials in Humans

**DOI:** 10.3390/nu14112263

**Published:** 2022-05-28

**Authors:** Marva Sweeney, Gracie Burns, Nora Sturgeon, Kim Mears, Kim Stote, Cynthia Blanton

**Affiliations:** 1Department of Biology, University of Prince Edward Island, Charlottetown, PE C1A 4P3, Canada; geburns@upei.ca; 2Department of Applied Human Sciences, University of Prince Edward Island, Charlottetown, PE C1A 4P3, Canada; nesturgeon@upei.ca; 3Robertson Library, University of Prince Edward Island, Charlottetown, PE C1A 4P3, Canada; kmears@upei.ca; 4Department of Allied Health Sciences, State University of New York, Empire State College, Saratoga Springs, NY 12866, USA; kim.stote@esc.edu; 5Department of Nutrition and Dietetics, Idaho State University, Pocatello, ID 83201, USA; cynthiablanton@isu.edu

**Keywords:** berries, polyphenols, blood pressure, intestinal microbiota, randomized controlled trials

## Abstract

Berry consumption has beneficial effects on blood pressure. Intestinal microbiota transform berry phytochemicals into more bioactive forms. Thus, we performed a systematic review of randomized clinical trials to determine whether berry polyphenols in foods, extracts or supplements have effects on both the profile of gut microbiota and systolic and diastolic blood pressure in humans. PubMed, Cochrane Library, Scopus, and CAB Abstracts (EBSCOhost) were searched for randomized clinical trials in humans published from 1 January 2011 to 29 October 2021. Search results were imported into Covidence for screening and data extraction by two blinded reviewers, who also performed bias assessment independently. The literature search identified 216 publications; after duplicates were removed, 168 publications were screened with 12 full-text publications assessed for eligibility. Ultimately three randomized clinical trials in humans met the eligibility criteria. One randomized clinical trial showed a low risk of bias while the other two randomized clinical trials included low, high or unclear risk of bias. Together the randomized clinical trials showed that berry consumption (Aronia berry, strawberries, raspberries, cloudberries and bilberries) for 8–12 weeks had no significant effect on both blood pressure and the gut microbiota. More randomized clinical trials are needed to determine the effects of berry consumption on the profile of gut microbiota and blood pressure in humans.

## 1. Introduction

The consumption of berry products is associated with reduced blood pressure [1,2,3]. Observational, interventional and in vitro studies have informed the identification of the berry components that protect against high blood pressure and the understanding of underlying mechanisms. The beneficial effects of berries are linked to their high concentration of bioactive polyphenols, particularly flavonoids such as anthocyanins [4,5]. Berries are an important focus of investigation regarding polyphenols and health outcomes partly because they are the dominant dietary source of anthocyanins in U.S. adults [6].

Large prospective cohort studies support the association of the intake of flavonoid-rich foods such as berries and improved blood pressure. An investigation of habitual flavonoid intake in cohorts of the Nurses’ Health Study and Health Professionals Follow-Up Study found that the highest intakes of anthocyanin, primarily from berries, was associated with an 8% reduction in incident hypertension [7]. Similarly, a study in the TwinsUK registry showed that higher habitual consumption of anthocyanins and berries was associated with significantly lower central systolic blood pressure and pulse wave velocity (a measure of arterial stiffness) [8].

Intervention trials have tested the impact of berry consumption on improved blood pressure and many, but not all, provide evidence suggestive of a causal relationship. A 12-week berry intervention showed significant reductions in systolic blood pressure and blood pressure variability, a predictor of coronary events [9], in high-normal and hypertensive men [10]. Similar berry feeding trials have demonstrated similar decreases in systolic blood pressure, particularly in subjects with the highest blood pressure measures [11,12,13,14]. Moreover, a meta-analysis of clinical trials examining effects of food sources of anthocyanins on cardiometabolic outcomes concluded that blood pressure was significantly reduced by consumption of anthocyanins from berries [15]. The scientific literature is not conclusive, however, as some intervention studies have shown inconsistent or no significant effects of berry consumption of blood pressure [16,17].

Mechanisms explaining the beneficial effects of berry consumption on blood pressure include modulation of the activity of vasoactive molecules. Berry extracts have been shown to cause dose-dependent, nitric oxide-related vasorelaxation and protection against damage from reactive oxygen species in isolated coronary arterial rings [18]. Importantly, an increased potency of microbial anthocyanin metabolites versus parent compounds in producing cardioprotective effects is observed [19]. Inhibition of angiotensin-converting enzyme, which creates the vasoconstrictor angiotensin II, by anthocyanins is another possible mechanism underlying the blood pressure-lowering activity of dietary berries [20].

As findings have accumulated on the beneficial effects of berry consumption on blood pressure, evidence increasingly supports a mediating role of the intestinal microbiota [21]. Specifically, intestinal bacteria transform dietary phenolic compounds to metabolites with greater bioactivity than the parent molecule [22]. Fruit phenolics also modify the intestinal microbiota community profile, which subsequently affects metabolite production [23,24,25,26]. The findings of a protective effect of berries against high blood pressure are complemented by evidence supporting a mediating role of the intestinal microbiota. A recent cross-sectional study demonstrated that higher intakes of total flavonoids and berries were associated with significantly lower systolic blood pressure [27]. Further, higher intakes of berries were associated with a more diverse and modified microbiota population that explained 11.6% of the relationship between the consumption of berries and blood pressure.

Given the potentially integral role of the intestinal microbiota in mediating the impact of dietary berries on blood pressure, we performed a systematic review of randomized controlled trials examining the interplay of dietary berries, the intestinal microbiota and blood pressure. While others have recently reported on the relationship between anthocyanins and the microbiota [28] and berry intake and blood pressure [29], this investigation is the first known to integrate these three factors (dietary berries, intestinal microbiota and blood pressure) in a systematic review of the literature.

## 2. Materials and Methods

This review was registered on PROSPERO (CRD42020186414) on 22 November 2021. The review was conducted according to the 2020 Preferred Reporting Items for Systematic Reviews and Meta-Analyses (PRISMA) guidelines [30]. The search strategy is reported according to the Preferred Reporting Items for Systematic reviews and Meta-Analyses literature search extension (PRISMA-S) [31].

### 2.1. Information Sources

The searches were conducted on 29 October 2021 in the following electronic databases: PubMed, Cochrane Library, Scopus, and CAB Abstracts (EBSCOhost).

### 2.2. Search Strategy

An initial list of search terms was developed by a health sciences librarian for the 4 concepts explored in this review: (1) berries/polyphenols; (2) gut microbiota; (3) blood pressure; and (4) randomized trials. The finalized list of terms was translated to PubMed by a second health sciences librarian, including the Cochrane Collaboration’s highly sensitive filter for identifying randomised trials. Bond University’s IEBH Systematic Review Accelerator tool (https://sr-accelerator.com/#/polyglottranslate, accessed on 1 October 2021) was used to translate the searches [32]. Formal peer review of the search strategies did not occur. All searches are available at https://doi.org/10.11571/upei-roblib-data/researchdata:766 (accessed on 7 February 2021). The search identified human studies published from 1 January 2011 to 29 October 2021.

### 2.3. Inclusion and Exclusion Criteria

The review included only human randomized clinical trials that evaluated the impact of berry polyphenols on gut microbiota and systolic and diastolic blood pressure. The dietary sources of berry polyphenols included foods, extracts or supplements. Participants in the trials included all ages, sexes, ethnicities and health/disease statuses. Exclusion criteria included in vitro studies, animal studies and observational studies.

### 2.4. Data Extraction

Literature search results were imported into Covidence, an online systematic review software for screening and data extraction. De-duplication occurred within Covidence as citations were imported into the software. Abstract and title screening, full-text review, data extraction, and bias assessment were completed independently by two blinded reviewers (GB, NS). Conflicts were resolved by review and discussion with all review authors. Extracted information is detailed in Table 1.

### 2.5. Quality Assessment

The assessment of risk of bias was conducting using the default assessment criteria in Covidence. This tool uses the Cochrane risk of bias domains including: sequence generation, allocation concealment, blinding of participants and personnel, blinding of outcome assessment, incomplete outcome data, selective reporting and other sources of bias [33]. The initial assessment was completed by two blinded reviewers (GB, NS). Conflicts were resolved by review and discussion with all review authors.

## 3. Results

The literature search identified 216 publications that were imported from screening of which 48 duplicates were removed, 168 publications were screened with 156 publications determined to be irrelevant. Ultimately, 12 full-text publications were assessed for eligibility with 9 out of the 12 publications excluded primarily due to lack of dietary berry polyphenol interventions and not including primary outcome measures of the gut microbiota or blood pressure. A total of three randomized clinical trials met the eligibility criteria [34,35,36]. The study flow diagram is shown in Figure 1.

Berry consumption had no significant effect on both blood pressure and the gut microbiota in the randomized clinical trials relevant to the systematic review topic of interest [34,35,36]. Istas et al. [34] showed Aronia berry consumption altered the gut microbiome with no effect on blood pressure. Of the three studies, the sample size of participants varied from 32 to 142, with an age range of 18 years to 75 years. All but one study included both male and female participants with differing baseline health status. Berry interventions were *Aronia melanocarpa* (black chokeberry), *Fragaria ananassa,* (strawberry), *Rubus idaeus* (raspberries), *Rubus* *chamaemorus* (cloudberries) and *Vaccinium myrtillus* (bilberry) with varying amounts of polyphenols. Placebo controls were used in two studies with one study providing generalized dietary advice to limit berry consumption. The duration of the study interventions ranged from 8 weeks to 12 weeks (Table 2). The studies were conducted in England, Finland, and Sweden.

The Cochrane risk of bias assessment was used in grading the strength of the body of evidence. Of the three randomized clinical trials, one study showed a low risk of bias and the other two studies included low, high or unclear risk of bias (Table 3).

## 4. Discussion

Berry fruits are a rich source of phytonutrients, especially polyphenolic flavonoids and condensed tannins [4,5]. We [14,37] and others [7,8,9,10,11,12,13,15] have shown that feeding berries improves many markers of cardiovascular disease (CVD) such as lowering blood pressures and improving diabetes. Observational, interventional and in vitro studies have suggested that flavonoids such as anthocyanins are responsible for the beneficial effect of berries on CVD [4,5]. Evidence increasingly supports a collaborative role for intestinal microbiota as they may transform flavonoids and other dietary phenolic compounds to metabolites with greater bioactivity than the parent molecule [21,22]. Indeed large polyphenols such as ellagitannins are not absorbed by human intestines unless they are transformed by gut microbes to bioavailable forms [38]. Therefore, this systematic review sought to answer the question “What are the effects of berry polyphenols on the profile of gut microbiota and subsequent polyphenol-mediated effects on blood pressure in humans?” Twelve full-text publications were screened and nine were subsequently excluded, so only three clinical trials met our inclusion criteria of measuring both blood pressure and intestinal microbial diversity in randomized control trials of human subjects [34,35,36], although an observational study provides valuable information to our question [27].

The three publications that met the criteria evaluated many different berry interventions in different forms and dosages [34,35,36]. The berries were *Aronia melanocarpa*, strawberry, raspberry, cloudberry and fermented bilberry, with no overlap of treatment among the three studies. The form of berries differed greatly, from whole fruit to purées, powders or beverages. Some studies included healthy individuals, while others evaluated berry effects on subjects at serious CVD risk. A summary of results from our review is shown in Table 2. In short, the results of this review found that these varied berry interventions at the doses and treatment durations had no effect on blood pressure or fecal microbial diversity. However, more research is needed to clarify a few trends that were seen in these studies, especially since a recent observational study concluded that there is a strong association between dietary berries and both microbiota and blood pressure-lowering [27].

Polyphenols have potent antioxidant, vasorelaxant, and anti-inflammatory actions. In one of the studies reviewed here [34], healthy male volunteers aged 18–45 y in London and the surrounding area were recruited into the double-blind, parallel randomized controlled trials. They were fed *Aronia melanocarpa* whole berries or powder, containing various polyphenols, with the largest percentage composed of the flavonol quercetin and its glycosides (29% of total polyphenols) and chlorogenic acids (26% of total phenolics). Although there were no effects on blood pressure, one of the outcomes evaluated in this systematic review, both *Aronia* treatments (whole fruits and powder) improved vascular endothelial function, where flow-mediated dilation improved by ~1–2% [34]. A 1% improvement has been associated with a decrease of 8–10% in CVD risk [39]. Thus, *Aronia* consumption has the potential to maintain cardiovascular health in healthy individuals. Furthermore, this same study found a positive correlation between *Aronia*-induced changes in abundance of some gut microbes (*Dialister*, *Phascolarctobacterium* and *Roseburia*) and beneficial effects of *Aronia* extract on flow-mediated dilation [34]. Together, this study reinforces the notion that the intestinal microbiota composition may affect the impact of berry consumption on markers of CVD.

Two other studies met inclusion criteria in this systematic review [35,36]. They evaluated the effect of berry feeding on individuals with CVD, specifically metabolic syndrome [35] and hypertension [36], and who were on various medications. Like Istas et al. [34], these papers also found no statistically significant effects on blood pressures or fecal microbiota diversity, although there was a trend to lower systolic blood pressure after eating a mixture of strawberries, raspberries and cloudberries for 8 weeks [35]. Systolic blood pressure fell by ~5 mmHg over the study [35], enough to reduce one’s risk for major cardiovascular events by ~10% according to a recent meta-analysis by the Blood Pressure Lowering Treatment Trialists’ Collaboration [40]. The intervention in that study contained primarily ellagitannins (~800 mg daily), with much lower amounts of anthocyanins (70 mg) and flavonols (4.1 mg). By comparison, Istas et al., fed healthy individuals 3 mg flavonols in whole berries and 35 mg flavonols in *Aronia* berry extract [34]. Previously anthocyanin intake has been linked to lowering the risk of hypertension [7]. It was interesting, and unexpected, that bilberries had no effect on blood pressures (36). Bilberries as European blueberries (*Vaccinium myrtillus*) are rich in anthocyanins and closely related to the North American blueberries (*Vaccinium angustifolium*) that we have studied. Our animal work shows that feeding blueberries to rats lowers systolic blood pressure [41,42] and 2 human randomized clinical trials showed trends to lowering blood pressures [14,37]. A systematic review published in 2012 concluded that there is a paucity of clinical trials evaluating blueberry consumption in humans at risk for disease [43]. Some researchers have shown that blueberries fed to humans lowers blood pressures [12], while others have not [17]. The discrepancies commonly observed are likely dependent on many factors such as the health status of the test subjects, the different treatment types (blueberry formulation and dosages), intervention durations, as well as inter-individual variations. Indeed, while diets rich in fruits and vegetables have known benefits on CVD, it is likely that individual components interact to produce the noted effects. Not all phytochemicals are absorbed equally, so bioavailability and bioactivity can vary depending on many factors.

A recent human study that evaluated the relationship between berries and both blood pressure and colonic microflora was excluded from our systematic review since it was observational in nature [27]. However, the results are important to note for this discussion. There was a positive correlation between higher intake of berries and specific flavonoids, notably anthocyanins, flavonols and tannins (proanthocyanidins), and lower systolic blood pressure [27]. Notably, a portion of the cardioprotective effects (~12%) were attributable to changes in the gut microflora, specifically increases in alpha and beta diversity, and decreases in *Parabacteroides*. It is important to emphasize that large inter- and intra-individual variations exist in the human fecal microflora composition [44], which is one reason why ongoing, larger and adequately powered randomized clinical trials are needed. Sex is another factor that can influence gut microbiota, potentially acting as a confounder in studies, with a need to further investigate how gut flora varies between sexes [45]. Istas et al. only studied men [34]. There was also variability in sample sizes and study durations, which could account for some of the heterogeneity in results.

The intestinal microbiota is investigated using a variety of approaches [46]. Amplicon sequencing of bacterial 16S rRNA gene is used to identify distinct bacterial taxons, usually at the genus level, within a sample. Shotgun meta-genomics, in contrast, characterizes the entire genetic composition of the sample rather than targeting a genomic sequence. This provides greater resolution of the bacterial community at the strain and species levels [47]. Reference databases are used to classify bacterial groups [48]. Another approach to microbial characterization is transcriptomics, which assesses the genes expressed by a microbial community. This method enables identification of active gene transcription and potential translation to proteins. Proteomics goes beyond transcriptomics to describe the protein products of microbes. Knowing the proteins produced by a bacterial community allows for identification of bacteria types and their functional capacity. For example, the identification of specific enzymes produced by a bacterial community is useful for determining the impact of diet on the microbiota and its metabolic potential, which influence nutrient bioavailability and health outcomes in the host [49,50]. Lastly, metabolomics evaluates the metabolites present in a biological sample. Measuring metabolites and their concentrations in blood, feces and urine can be used to describe how the microbiota responds to an exposure, such as diet and exercise, and the bioactive compounds produced [51,52].

While berry polyphenols are well tolerated and considered safe in healthy people, contraindications to their use are important to note [53]. Polyphenols found in berries are shown to modulate cytochrome P450 (CYP) enzymes in intestine and liver [54,55,56,57] and therefore the concurrent intake of polyphenol-rich foods/extracts and specific medications poses risks of adverse effects. For example, a preparation of aronia berry was shown to induce toxicity resulting in rhabdomyolysis in a patient undergoing cancer chemotherapy with trabectedin [58]. The authors concluded that the aronia product inhibited CYP enzymes that metabolize trabectedin, leading to toxic levels of the drug. Further, high intakes of quercetin have been found to interact with levels of digoxin, a commonly used cardiac medication with a narrow therapeutic range, and simultaneous administration in experimental animals resulted in death [59]. Polyphenol–drug interactions have been a focus of study for decades but recent research demonstrates a complex system of polyphenol-protein interactions in the body that has been mapped [60].

There are strengths and limitations of our systematic review. All screening, text review, data extraction, and bias assessment was completed independently by two blinded reviewers, with conflicts resolved by all authors in a process determined prior to beginning. Our conclusions are only as valid as the studies that met inclusion criteria. Any issues with the design and execution of randomized clinical trials will raise questions about the validity of the findings. One study scored consistently low in terms of risk of bias on the Cochrane scale [34], whereas the other two studies had some elements of high bias risk, specifically relating to blinding of the participants or experimenters (Table 3). All three studies had a parallel design and were done in Europe over several weeks (8–12-week interventions), but one study excluded female subjects [34], and one study lacked a placebo control [35]. The sample size and age range of participants varied dramatically, and participants had differing baseline health statuses. Further rigor is warranted, and more research is needed to clarify our question of whether oral intake of berries impacts gut microbiota and blood pressure.

## 5. Conclusions

Berry fruits are rich in flavonoids, phytochemicals that are metabolized by intestinal microbiota into more bioavailable and bioactive compounds. Diets rich in berries have been shown to lower blood pressure, benefiting CVD. This systematic review asked whether dietary berry interventions would alter both blood pressure and colonic microflora. Our results indicate that only three published randomized clinical trials addressed this question. There was no evidence that dietary berry interventions such as strawberries and bilberries benefitted both blood pressure and composition of the intestinal microbiome, although there were trends to cardioprotection in these studies. A more recent observational study evaluating human berry consumption prospectively did show a positive correlation between intake of berries, as well as flavonoids, lower blood pressure and microbiome composition. In fact, ~12% of the cardioprotection was attributed to changes in the gut microflora. We conclude that more studies are needed to answer the question, given the huge diversity of both microbes and berry flavonoids.

## Figures and Tables

**Figure 1 nutrients-14-02263-f001:**
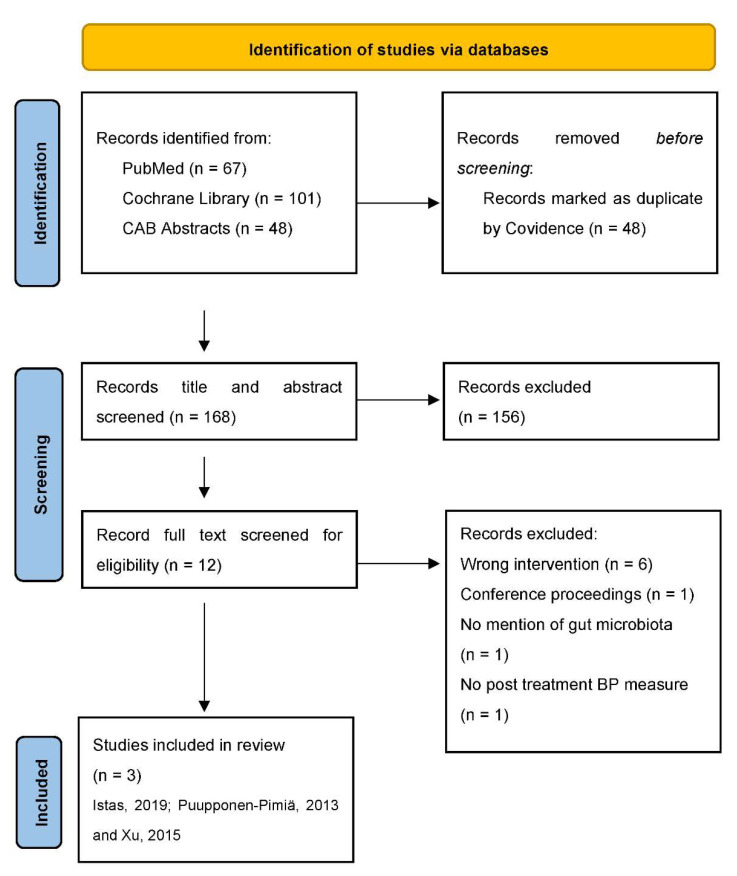
PRISMA diagram: flow of studies through the review. Abbreviation: PRISMA, Preferred Reporting Items for Systematic Reviews and Meta-Analyses.

**Table 1 nutrients-14-02263-t001:** Effects of dietary berry polyphenols on blood pressure and the intestinal microbiota.

Category	Data		
**General Information**	Title	Lead author contact details	
	Year of publication	Study region	
	Journal name	IRB/REB approval	
	Registered clinical trial number		
**Characteristics of included studies**	Blinding Methods Aim of study Randomization Study type Washout period for crossover design Duration of study Start date End date Study funding sources Study funding sources Possible Conflicts of interest for study authors Participants	Population description Sex Inclusion criteria Exclusion criteria Method of recruitment of participants Total number of participants (randomized at baseline) Dietary treatment 1 Dietary treatment 2 Form of control Dose of control Baseline health	
**Baseline Population Characteristics**	Control Age Sex Race Systolic BP (peripheral) Diastolic BP (peripheral) Systolic BP (central) Systolic BP (central) Diastolic BP (central) BMI Number of Participants	Treatment 1 Age Sex Race Systolic BP (peripheral) Diastolic BP (peripheral) Systolic BP (central) Systolic BP (central) Diastolic BP (central) BMI Number of Participants	Treatment 2 Age Sex Race Systolic BP (peripheral) Diastolic BP (peripheral) Systolic BP (central) Systolic BP (central) Diastolic BP (central) BMI Number of Participants
**Intervention and Comparisons**	Type of Biologicals Samples Time Point of Sample Collection (Military Time)		
**Outcomes**	Outcome—Time Point 1 Outcome—Time Point 2 Outcome—Time Point 3 Impact on microbial community	Gut microflora Study dropout rate (number and percentage) Power calculation for sample size determination Statistical analysis Statistical design appropriate	Other comments

IRB/REB, Institutional Review Board/Research Ethics Board; BP, blood pressure; and BMI, body mass index.

**Table 2 nutrients-14-02263-t002:** Effects of dietary berry polyphenols on blood pressure and the intestinal microbiota.

	Istas, 2019 [34]	Puupponen-Pimiä, 2013 [35]	Xu, 2015 [36]	
***n* of Participants (M/F)**	66 (66/0)	32 (13/19)		142 (92/50)
**Participant age**	23 ± 4 years	51 ± 7 years		66 ± 4 years
**Health status**	Healthy	Metabolic syndrome		Hypertension
**Intervention**	*Aronia melanocarpa*: “aronia whole fruit” capsules (12 mg polyphenols, 10 g berries) or *Aronia melanocarpa*: “aronia extract” capsules (116 mg polyphenols, 75 g berries)	300 g fresh berries (100 g strawberry purée, 100 g frozen raspberries, 100 g frozen cloudberries) substituting other sources of carbohydrates normally consumed		Fruit drink (10 g fresh bilberries fermented with *L. plantarum* strain DSM 15313 with placebo probiotic powder) or fruit drink with probiotic powder, *L. plantarum* strain DSM 15313 (without bilberries)
**Control**	Control capsules, matched in appearance to both intervention capsules, contains maltodextrin and no polyphenols	Usual diet with berry restriction		Placebo drink plus placebo probiotic powder (without bilberries or *L. plantarum* strain DSM 15313)
**Study design**	Parallel (3 groups)	Parallel (2 groups)		Parallel (3 groups)
**Duration**	12 week intervention	16 week (4 week run in with berry restriction, 8 week intervention, 4 week recovery period)		14 week (2 week run in period, 12 week intervention)
**Impact on blood pressure**	Change in blood pressure at 12 weeks (mmHg) Aronia whole fruit vs. control: PSBP −0.6 (−6.8, 5.6), PDBP −1.8 (−7.3, 3.7), CSBP −1.8 (−7.3, 3.8), CDBP −2.4 (−9.4, 4.6) Aronia extract vs. control: PSBP −0.4 (−6.7, 5.8), PDBP 0.8 (−4.8, 6.4), CSBP −1.1 (−6.7, 4.5), CDBP 1.2 (−5.8, 8.3)	Change in blood pressure at 12 weeks (mmHg) Berry: SBP −4.8 ± 5.9, DBP −2.8 ± 2.8 Control: SBP −8.2 ± 4.5, DBP −2.9 ± 3.1		Change in blood pressure at 12 weeks (mmHg) Bilberries fermented by *L. plantarum* strain DSM 15313: SBP 1 (−6, 8), DBP 2 (−2, 8) *L. plantarum* strain DSM 15313: SBP −1 (7.25, 7.25), DBP −3 (−7, 2) Placebo control: SBP 1.5 (−5, 7.75), DBP −1 (−4, 5.5)
**Impact on microbial community**	NC in fecal microbiota diversity Aronia whole fruit capsules: ↑ Bacteroides * Aronia extract capsules: ↑ Anaerostipes * Placebo capsules: ↑ Clostridium XiVb *	NC in fecal microbiota Diversity		NC in oral or fecal microbiota diversity

Arrows indicate reported changes ↑, increased; NC, no statistically significant changes between baseline and end of intervention duration; PSBP, peripheral systolic blood pressure; PDBP, peripheral diastolic blood pressure; CSBP, central systolic blood pressure; CDBP, central diastolic blood pressure; SBP, systolic blood pressure; DBP, diastolic blood pressure; M/F: Male/Female; * *p* < 0.05.

**Table 3 nutrients-14-02263-t003:** Cochrane assessment of risk of bias.

Author, Year	Sequence Generation ^1^	Allocation Concealment ^2^	Blinding of Participant and Personnel ^3^	Blinding of Outcome Assessor ^4^	Incomplete Outcome Data ^5^	Selective Outcome Reporting ^6^	Other Sources of Bias ^7^
Istas, 2019 [34]	Low risk	Low risk	Low risk	Low risk	Low risk	Low risk	Low risk
Puupponen-Pimiä, 2013 [35]	Low risk	Unclear	High Risk	Unclear	Low risk	Low risk	Unclear
Xu, 2015 [36]	Low risk	Low risk	High Risk	Unclear	Low risk	High risk	Unclear

^1^ Describe the method used to generate the allocation sequence in sufficient detail to allow an assessment of whether it should produce comparable groups. ^2^ Describe the method used to conceal the allocation sequence in sufficient detail to determine whether intervention allocations could have been foreseen in advance of, or during, enrollment. ^3^ Describe all measures used, if any, to blind study participants and personnel from knowledge of which intervention a participant received. Provide any information relating to whether the intended blinding was effective. ^4^ Describe all measures used, if any, to blind outcome assessors from knowledge of which intervention a participant received. Provide any information relating to whether the intended blinding was effective. ^5^ Describe the completeness of outcome data for each main outcome, including attrition and exclusions from the analysis. State whether attrition and exclusions were reported, the numbers in each intervention group (compared with total randomized participants), reasons for attrition/exclusions where reported, and any re-inclusions in analyses performed by the review authors. ^6^ State how the possibility of selective outcome reporting was examined by the review authors, and what was found. ^7^ State any important concerns about bias not addressed in the other domains in the tool. If particular questions/entries were pre-specified in the review’s protocol, responses should be provided for each question/entry.

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
