# Peer review of "The Effects of Berry Polyphenols on the Gut Microbiota and Blood Pressure: A Systematic Review of Randomized Clinical Trials in Humans"

_nutrients, 2022, doi:10.3390/nu14112263_

Round 1

Reviewer 1 Report

In the manuscript “The effects of berry polyphenols on the gut microbiota and blood pressure: a systematic review of randomized clinical trials in humans” by Marva Sweeney et al., the authors performed an accurate search in different databases to select studies aimed to evaluate the effect of berry polyphenols on gut microbiota and blood pressure.

They accurately describe the methodological approach they used to carry on the study. This approach finally led to the selection of only three clinical trials, that are not a lot.

The authors say that no statistically significant effects on blood pressures or fecal microbiota diversity due to the assumption of berries was found.

The authors admit that their review has some limits due to different factors, including the lack of placebo controls in one of the studies, the presence of only male subjects in another study, the great variation in sample size and age of participants, the different baseline health status of participants, etc..

Thus even though the idea and the approach are potentially good, the conclusions are rather vague and of little significance.

Reviewer 2 Report

This systematic review covers interesting and emerging topic of the effects of berry polyphenols on the gut microbiota and blood pressure. Authors did a great job in breifing out articles they studied (L50-L69). They also perfectly outlined possible mechanisms by wich berry consumption has beneficial effects on blood preassure (L71-L94). Modern recomendations (L104-L109) and clear statement of strengths and limitations of their work (L322-L334) gained extra value to this manuscript. 

Minor corrections and editions throughout the text together with nicely descibed conclusion could throttle future research and publications on the described topic.

L34: First sentence (statement) goes without a reference. Please add.
L121: Get rid of underline (after link)
L143 (table # and description). It is confusing to have title of the table above The Table and description - separately, below. Please merge it with L144 and L145 underneath The Table (like you did for figures).
L198 + L199/L200 - same as previous
Discussion: while topic of microbiota changes throughout different studies was widelly covered, it would be beneficial for readers to preliminarily touch methods by wich microbiota is studied; i.e., metaproteonomics (PMCID: PMC8147213), metagenomics (PMCID: PMC9026403), trancriptomics (doi.org/10.3390/app12052483), etc.

Otherwise, I would like to greet authors with a nicely written manuscript and wish them further success.

Reviewer 3 Report

Dear Editor,

I carefully read the manuscript by Sweeney et al.

My comments and suggestions for the authors are the following:

  • The background paragraph needs to be carefully revised. Currently, it has been written in a non formal way. Moreover, this paragraph needs to be shortened (now, the manuscript is overall unbalanced).
  • Line 112: I suggest the authors to include also Google Scholar among the searched database.
  • Table 1 - "Gender" should be replaced by "sex".
  • The authors should more deeply discuss their findings.
  • Overall, the article is neither original nor interesting and it does not bring new information.

Round 2

Reviewer 1 Report

The authors responded satisfactorily to the comments. I believe that in the present form the manuscript can be suitable for publication.

Author Response

Thank you for your comments. 

Reviewer 3 Report

Dear Editor,

I carefully read the revised version of the manuscript that is improved in comparison with the previous one. I have some other comments and  suggestions for the authors:

 - The manuscript is not balanced in its parts. In particular, the "Background" paragraph should be shortened

 - The references should be updated. In particular, the authors should refer to doi: 10.1007/s40292-018-0296-6, doi: 10.1007/s40292-021-00474-6 and doi: 10.1097/HJH.0000000000002353

Author Response

Thank you for your valuable suggestions, which we have applied to the manuscript.

1) The introduction (or background) section of the manuscript has been shortened and retains only the most relevant background information for providing context for this review’s topic.

2) The recommended references have been inserted in line 38.

Thank you for your reconsideration of our manuscript.